# GP130 Cytokines in Breast Cancer and Bone

**DOI:** 10.3390/cancers12020326

**Published:** 2020-01-31

**Authors:** Tolu Omokehinde, Rachelle W. Johnson

**Affiliations:** 1Program in Cancer Biology, Vanderbilt University, Nashville, TN 37232, USA; 2Vanderbilt Center for Bone Biology, Department of Medicine, Division of Clinical Pharmacology, Vanderbilt University Medical Center, Nashville, TN 37232, USA; 3Department of Medicine, Division of Clinical Pharmacology, Vanderbilt University Medical Center, Nashville, TN 37232, USA

**Keywords:** breast cancer, bone metastasis, glycoprotein 130 (gp130), interleukin-6 (IL-6), oncostatin M (OSM), leukemia inhibitory factor (LIF), ciliary neurotrophic factor (CNTF), cancer stem cells (CSC)

## Abstract

Breast cancer cells have a high predilection for skeletal homing, where they may either induce osteolytic bone destruction or enter a latency period in which they remain quiescent. Breast cancer cells produce and encounter autocrine and paracrine cytokine signals in the bone microenvironment, which can influence their behavior in multiple ways. For example, these signals can promote the survival and dormancy of bone-disseminated cancer cells or stimulate proliferation. The interleukin-6 (IL-6) cytokine family, defined by its use of the glycoprotein 130 (gp130) co-receptor, includes interleukin-11 (IL-11), leukemia inhibitory factor (LIF), oncostatin M (OSM), ciliary neurotrophic factor (CNTF), and cardiotrophin-1 (CT-1), among others. These cytokines are known to have overlapping pleiotropic functions in different cell types and are important for cross-talk between bone-resident cells. IL-6 cytokines have also been implicated in the progression and metastasis of breast, prostate, lung, and cervical cancer, highlighting the importance of these cytokines in the tumor–bone microenvironment. This review will describe the role of these cytokines in skeletal remodeling and cancer progression both within and outside of the bone microenvironment.

## 1. Introduction

Upon dissemination into the bone marrow, breast cancer cells and other tumor types encounter a rigid [1], hypoxic [2] microenvironment containing bone-resident immune and stromal cell populations. It is hypothesized that disseminated tumor cells (DTCs) compete for the hematopoietic stem cell niche, and are thus maintained in a quiescent state by interactions with osteoblast lineage cells [3] for an indefinite period of time. Bone-DTCs secrete factors (e.g., parathyroid-hormone-related protein (PTHrP)) that stimulate the receptor activator of NF-κB (RANK)-RANK ligand (RANKL) axis and promote osteoclastogenesis [4]. These factors may enable tumor cells to overcome quiescence [5], but it remains unclear whether some breast cancer cells begin secreting these factors prior to dissemination or during circulation, or whether the bone microenvironment induces breast cancer cells to stimulate osteoclasts. Increased osteoclastogenesis gives rise to localized bone resorption and the release of cytokines and growth factors from the bone matrix that stimulate tumor cell growth and further enhance the RANK-RANKL signaling cascade to promote bone resorption [4], resulting in overt tumor-induced bone disease.

Cytokines and cytokine receptors have a wide range of physiological functions and biological activities in many tissues and cell types [6]. The interleukin-6 (IL-6)/glycoprotein130 (gp130) cytokine family has been implicated not only in inflammation and immune response, but also in hematopoiesis, neuronal regeneration, bone remodeling, and cancer [7,8]. In this review, we will primarily focus on the role of the gp130 cytokines in cancer and bone.

The gp130 co-receptor is expressed in almost all major organs of the human body [9] and is the key signaling transducer that unites the IL-6 cytokine family. Each of the cytokines in the family binds to a cytokine-specific receptor and will complex with at least one subunit of gp130 to form its cell surface receptor complex. The targeted deletion of *Il6st* (the gp130 mouse gene) in mice resulted in embryonic lethality, with greatly reduced numbers of hematopoietic progenitors, impaired development of red blood cells, and defects in heart development [10]. *Il6st* null mice also exhibited poor bone development and a reduction in osteoblast number and function [11]. While the osteoclast number was increased with gp130 deletion [11,12], osteoclasts had poorly developed ruffled borders and the mice were slightly hypocalcemic, suggesting a defect in osteoclast activity. These data highlight the importance of gp130 in development, bone homeostasis, hematopoiesis, cell survival, and growth.

All of the IL-6 cytokines are dependent upon gp130 to induce downstream signaling pathways to affect a wide range of biological processes. When IL-6 binds to the IL-6 receptor (IL-6R), it triggers a homodimeric association with gp130 to form its receptor complex [13], allowing signal transduction to occur in the target cell. Similar results have been shown for interleukin-11 (IL-11) when binding to the IL-11 receptor (IL-11R), and other gp130 family members induce the recruitment of cytokine-specific receptor chains [14]. An example of this is the leukemia inhibitory factor (LIF) receptor (LIFR), which is required for signal transduction induced by the ligands LIF, cardiotrophin-1 (CT-1), and ciliary neurotrophic factor (CNTF). LIF signals by first binding to its cytokine-specific receptor LIFR and then recruits gp130, forming a heterodimeric receptor complex. CT-1 also signals by binding to LIFR and inducing heterodimerization with gp130, but there is evidence of a third receptor involved in signaling for CT-1, forming a possible heterotrimeric receptor complex [15]. Signal transduction for CNTF requires that it binds to the CNTF receptor (CNTFR) first, and then recruits LIFR and gp130, forming a heterotrimeric receptor complex. Oncostatin M (OSM) is unique because it can form two different heterodimeric receptor complexes, where OSM first binds to gp130, and then recruits either the OSM receptor (OSMR) or LIFR [16] (Figure 1). IL-27, which consists of IL-27p28 (p28) and Epstein-Barr virus induced 3 (EBI3), is known to signal through a receptor complex of WSX-1 (also referred to as interleukin 27 receptor subunit alpha) and gp130, in order to induce downstream signal transduction and the activation of STAT3 [8,17,18]. When IL-27p28 signals and forms complexes independent of EBI3, it is referred to as IL-30 [19].

Signal transduction through gp130 by any of the IL-6 family cytokines generally results in the activation of three major downstream pathways: the Janus-activated kinase (JAK)–signal transducer and activator of transcription (STAT) pathway, the Ras-Raf mitogen-activated protein kinase (MAPK, MEK/ERK) signaling cascade, and the phosphatidylinositol 3-kinase-dependent (PI3K/AKT) pathway [20,21,22,23]. The Hippo-Yes-associated protein (Hippo-YAP) pathway has also been shown to be negatively regulated downstream of LIFR [24]. However, in the osteoblast lineage, it has been shown that OSM activates distinct signaling pathways, depending upon whether it complexes with OSMR or LIFR [25], suggesting that these cytokines and their specific receptor complexes may induce specific downstream signals in bone-resident cells. A comprehensive comparison of the downstream pathways activated by the different cytokines after binding to breast cancer cells has not been conducted. Despite the similar sequence homology, structure, and intron-exon and promoter elements between OSM and LIF [26], the individual IL-6 cytokines have differing roles in cancer and bone biology. This may be partly due to tissue specificity for ligand and receptor expression or the activation of different downstream signals, which will be discussed in the following sections. 

## 2. gp130 in Physiological Bone Remodeling

Bone-resident osteoblasts (bone-forming cells), osteocytes (mechano-sensing terminally-differentiated osteoblasts), and osteoclasts (bone-resorbing cells) maintain bone homeostasis and health through the tightly regulated process of bone formation and bone resorption [27]. 

The gp130 cytokines are recognized as key regulators of bone remodeling. Il-6, Il-11, and Osm have been shown to promote bone formation by increasing alkaline phosphatase activity on mouse pre-osteoblast MC3T3 cells and primary mouse calvarial cells [28], and Osm [29], Ct-1 [30], Lif [31], and Il-6 [32] stimulate bone formation in vivo. Importantly, it has been demonstrated that Il-6 [33,34], Il-11 [35], Osm [29], Lif [36,37], and Cntf [38,39] are expressed by osteoblast-lineage cells and that Ct-1 is expressed by osteoclasts [30], suggesting that tumor cells in the bone marrow will encounter these signals in the physiological bone marrow microenvironment (Figure 2). Many of these factors are also expressed in skeletal muscle [40,41,42], suggesting that they may act in a paracrine manner on the adjacent bone.

The mechanisms by which these cytokines induce bone formation vary. Osm, in addition to Lif and Ct-1, can complex with Lifr/gp130 to inhibit the production of sclerostin—a potent inhibitor of bone formation [43]—in late differentiated osteoblasts and osteocytes. Osm can also act through Osmr/gp130 to promote osteoblast differentiation and increase Rankl production [29]. Therefore, Osm acting through Lifr/gp130 results in increased Wnt signaling and bone formation, while Osm acting through Osmr/gp130 increases osteoclastogenesis. In vitro and in vivo evidence suggests that this is due to the differential activation of STAT3 signaling over STAT1 signaling [25]. In physiological bone remodeling, mouse genetic knockout studies of the receptors have revealed the importance of these cytokines in physiological bone remodeling. Mice deficient for gp130 [11] and Lifr [44] have increased osteoclast numbers, low trabecular bone mass, various bone abnormalities, and impaired bone formation. The knockout of Lif [45], and Ct-1 [30] gave rise to a similar phenotype, with impaired osteoblast function and large osteoclasts in both neonate and adult mice. In contrast, the deletion of Osmr [29] and Il-11R [46] resulted in suppressed osteoclast differentiation, and a high trabecular bone volume, number, and thickness. The observed increase in bone volume in Osmr knockout mice is likely due to its negative regulation of sclerostin [29,47]. The mechanism underlying the increased bone volume in Il-11r knockout mice remains unclear.

Il-6 [48,49,50], Il-11 [48,51], Lif [48,49,50], Ct-1 [30,49], and Osm [48,49,50] are also known to induce pro-osteoclast effects by acting on osteoblast lineage cells to produce Rankl. Il-6, Lif, Osm, and Ct-1 induce the formation of tartrate-resistant acid phosphatase (TRAP)-positive multinucleated cells (MNC) and enhance osteoclast activity in vitro [48,49]. However, the role of these cytokines in vivo shows cytokine-specific phenotypic variations [52]. Gp130 null mice were observed to have very high osteoclast numbers, but also embryonic and hematopoietic defects [11]. The conditional knockout of gp130 in late osteoblasts and osteocytes [47], as well as osteoclasts [53], resulted in reduced osteoblast numbers and bone formation, but no change in osteoclasts. Similar to gp130 deletion, the genetic deletion of Ct-1, Lif, and Lifr also produced an increase in osteoclast formation. Ct-1 null mice had increased osteoclast formation and many large osteoclasts, but with abnormalities in their function, making the bones abnormally dense [30]. The knockout of Lif or Lifr resulted in an increase in large osteoclasts, with activity clustered near the growth plate in young mice [44,45]. 

These cytokines orchestrate bone remodeling to maintain bone homeostasis; however, bone-DTCs can hijack bone remodeling to alter the environment and make a more suitable environment for tumors to grow. Breast cancer cells induce osteolytic destruction to support their own growth and survival. By expressing and releasing cytokines such as IL-6 and IL-11, they promote their own growth through autocrine signaling and stimulate osteoclastic bone resorption through paracrine signaling. Importantly, breast cancer cells can also respond to these cytokines produced by the bone-resident cells since they express gp130 and many of the cytokine-specific receptors [54,55,56]. In this regard, understanding the role for gp130 cytokines in normal bone remodeling is essential to understanding the impact of gp130 cytokines and signaling in bone-DTCs. 

## 3. Tumor Niches within the Bone

Breast cancer cells frequently metastasize to distant organs, including the brain, lung, and bone marrow. Through mechanisms that remain unclear, some breast cancer cells will enter a period of quiescence in which they reside as non-proliferative, dormant tumor cells in the bone marrow [57]. It has been proposed that bone-DTCs interact with bone-resident cells and compete with hematopoietic stem cells (HSCs) in different niches [3,58] that are present in the bone marrow, including the perivascular niche and the endosteal niche. The perivascular niche, which can be found along the surface of the vasculature throughout the bone marrow, contains blood vessel-lining endothelial cells and mesenchymal stem cells (MSCs) that support HSCs by expressing factors that promote their growth and maintain their populations [59,60,61,62]. Factors such as stem cell factor (SCF) [59], CXC chemokine ligand 12 (CXCL12) [63,64], Notch signaling components [65], and E-selectin [66] have all been shown to help maintain HSCs in the perivascular niche. In contrast, deletion of the co-receptor gp130 in hematopoietic and endothelial cells resulted in bone marrow dysfunction and reduced hematopoietic cells in mice [67], suggesting that gp130 signaling is important for maintenance of the hematopoietic niche. The endosteal niche, which is localized to both the trabecular and endocortical bone surfaces, is rich in bone-resident cells, including osteoblasts and bone-lining cells. Osteoblast-linage cells have been observed to interact with HSCs through adherens junction molecules N-cadherin and β-catenin to regulate the HSC population, including quiescent/long-term HSCs [68,69]. This was demonstrated when HSCs that adhered to osteoblast-lineage cells were able to retain 5-bromodeoxyuridine (BrdU) for long periods of time [69]. In addition to adherens junction molecules, the interaction between Tie2, a receptor tyrosine kinase expressed on HSCs, and its ligand, Angiopoietin-1 (ANGPT1), produced by osteoblasts, promotes the tight adhesion of HSCs to osteoblasts and promotes the quiescence of HSCs both in vitro and in vivo [70]. Since vasculature can be found throughout the bone marrow, these niches can overlap in the bone and may simultaneously influence the HSC compartment in the bone marrow. When bone-DTCs traffic to the bone marrow, they are able to take advantage of these bone marrow niches to survive and persist in the bone for long periods of time. An example of this can be seen where stable microvasculature from the perivascular niche maintains tumor dormancy in breast cancer cells through endothelial cells expressing thrombospondin-1 (THBS1) [71], an adhesive glycoprotein that mediates cell-to-cell and cell-to-matrix interactions and a known inhibitor of neovascularization and tumorigenesis. This effect is negated by sprouting endothelial tips and neovascularization, which is characterized by reduced THBS1 expression and increased expression of transforming growth factor beta 1 (TGF-β1) and periostin (POSTN). 

In addition, osteoblasts and other osteogenic cells that promote HSC maintenance have been shown to contribute to the quiescence of tumor cells [3,72]. One group has shown that prostate cancer cells target the endosteal niche during metastasis and outcompete the HSCs in the bone marrow by competitively binding to osteoblasts and downregulating the expression of niche-adhesion molecules and self-renewal factors, such as NOTCH1, TIE2, BMI1, and INK4A [3]. While the contribution of the individual gp130 cytokines to tumor dormancy has not been fully resolved, LIFR has been shown to promote tumor dormancy [73] and function as a breast tumor suppressor [24,74]. The effects of individual cytokines in the context of breast cancer are highlighted below.

## 4. gp130 Cytokines in Breast Cancer 

Breast cancer is commonly categorized by hormone receptor expression and can be classified into distinct groups: estrogen receptor-positive (ER+), human epidermal growth factor receptor 2-positive (HER2+), progesterone receptor-positive (PR+), or triple-negative breast cancer (TNBC) [75]. Approximately 65%–75% of breast cancer cases are ER+, 25%–30% have the HER2 gene amplified, and 10%–20% of cases involve triple-negative breast cancer (TNBC), one of the most aggressive forms of the disease [75,76,77,78]. Similarly, breast cancer cell lines can also be distinguished by their hormone receptor expression, with phenotypes similar to the clinical counterpart (e.g., TNBC breast cancer cell lines are highly metastatic in mouse models, readily colonizing the lung or bone marrow after intravenous inoculation, while ER+ MCF7 cells do not readily colonize and exhibit slow or no growth in distant metastatic sites following inoculation [73,79]). 

In the context of the gp130 cytokines, IL-6 has been reported by numerous groups to play a role in breast cancer progression, and these effects correspond to hormone receptor expression. ER+ breast cancer patients tended to have lower levels of sIL-6R when compared to ER- patients, and increased levels of sIL-6R were associated with increased recurrence when compared to patients with lower levels of sIL-6R [80]. Interestingly, in silico modeling and in vitro testing of two selective estrogen receptor modulators (SERMs), raloxifene and bazedoxifene, revealed that they are able to bind to gp130, selectively downregulate IL-6-mediated STAT3 phosphorylation, and significantly inhibit STAT3 activity in ER- SUM159 breast cancer cells [81]. In contrast to ER+ breast cancer, both HER2+ and TNBC have elevated levels of IL-6, causing an autocrine feedback loop through IL-6-activated STAT3 [82,83]. The inhibition of IL-6 by the use of an IL-6 antagonist, tocilizumab, or through shRNA, resulted in decreased tumor growth, reduced cancer stem cells (CSCs), and the suppression of colony formation in HER2+ and TNBC studies [82,84]. 

While a large body of work has focused on the connection between the hormone receptor status and the IL-6/gp130 signaling axis, there have been few studies that have focused on LIF/LIFR and OSM/OSMR in connection with hormone receptor status. Dhingra et al. reported that LIFR expression in patient tumors was significantly correlated with the presence of estrogen receptor [85], and LIFR expression and function are typically the highest in ER+ breast cancer cell lines [73], although ER− SUM159 cells also possess an active LIFR capable of inducing downstream signals in response to the ligand [24,73]. Recent work by Li et al. reported that nuclear p21-activated kinase 4 (nPAK4) co-localized with endogenous ER-alpha (ERα) in the nucleus of ER+ MCF7 and ZR-75-30 breast cancer cells, resulting in the recruitment of the PAK4-ERα complex to estrogen response elements (EREs) upstream of the LIFR promoter, inhibiting the expression of LIFR and promoting bone metastasis [86]. In the context of OSM/OSMR signaling, high levels of OSM and OSMR mRNA expression were associated with low expression of ESR1 (ER) and ER-regulated genes in a breast cancer gene expression data set. That same study also noted that recombinant OSM potently suppressed the ER protein and mRNA expression in vitro and that loss of ER expression was necessary for OSM-mediated signal transduction and migratory effects in ER+ MCF7 and T47D breast cancer cells [55]. Overall, these studies did not show a correlative trend in regard to the hormone receptor status and the expression of gp130 cytokines, but these studies do suggest that ER may negatively regulate both LIFR and OSMR in breast cancer cells. Highlighted in the next section, we will discuss the often contradictory effects of some of the gp130 cytokines in relation to breast cancer, and how these correspond to the hormone receptor status. 

*IL-6:* Of all the gp130 cytokines, IL-6 is perhaps the most well-studied. Many groups since the 1980s have demonstrated that recombinant IL-6 slows proliferation of breast cancer cells in 2D cultures, with most of these studies focusing on ER+ human breast cancer cell lines like MCF7, T47D, and ZR-75-1 cells [87,88,89,90,91,92]. While there are a few reports indicating that IL-6 cytokines do not affect MCF7 tumor cell proliferation in vitro [93,94], these studies typically looked at early time points (e.g., 48–72 h of treatment). It is important to note that across all of these studies, the source of IL-6 was either not reported, came from different commercial vendors, or was produced in house. Of those that reported a commercial source for IL-6, none of these came from the same vendor. Therefore, while there is a large amount of variation in the extent to which IL-6 inhibits proliferation, the overwhelming body of evidence suggests that IL-6 inhibits breast cancer cell proliferation in 2D cultures. It is also worth noting that while IL-6 appears to inhibit proliferation, it was also found to promote the motility of MCF7, T47D, and ZR-75-1 cells [90,91], suggesting that it is not an entirely benign factor. IL-6 requires the expression of gp130 and the IL-6 receptor (either expressed on the cell surface or in a soluble form) in order to elicit intracellular signaling, but the vast majority of these studies did not examine the expression and contribution of the receptors [87,88,90,91,92]. Chiu et al. reported that IL-6 inhibits MCF7, T47D, and ZR-75-1 cell proliferation and determined that all three cell lines secrete a soluble IL-6 receptor and express gp130 [89], but did not test whether adding an additional soluble IL-6 receptor enhanced or changed this effect. In contrast, Jiang et al. reported no effect of IL-6 on MCF7 cell proliferation treated with either IL-6 or a soluble IL-6 receptor, but did not test the combination of IL-6 and a soluble IL-6 receptor or examine whether an IL-6 receptor or gp130 is expressed on MCF7 cells in their hands [94]. This may be of importance since the MCF7 cell line is notoriously heterogeneous in nature [95]. It is therefore possible that the effects of these cytokines on proliferation may be partly dependent on the expression and availability of the receptors, but this is difficult to say with certainty given that most studies have not examined receptor expression. 

It is also important to note that there is one study which suggests that IL-6 stimulates the proliferation of ER+ MCF7 and BT474 cells [96], and that this study had two notable differences from the aforementioned studies. First, the assay was conducted with a fluorescence reporter, in contrast to the thymidine incorporation and cell count studies that were previously used, and second, the cells were grown in a 3D tumor culture system. This key difference may reveal important differences in the effect of IL-6 on tumor cell proliferation in vitro and suggests that the ability of IL-6 to promote proliferation is dependent upon the environment of the tumor cell. The expression or secretion of an IL-6 receptor and gp130 was not evaluated in this study, so it is not clear whether tumor cells cultured in 3D systems expressed different levels of receptors compared to cells cultured in 2D systems. It is worth noting that the observed inconsistencies in IL-6-induced proliferation are not due to the cell line hormone receptor status, since the same ER+ cell lines were used across multiple studies.

In the context of bone, IL-6 is well-known to stimulate mesenchymal progenitor differentiation towards the osteoblast lineage, while also promoting RANKL expression in osteoblasts and osteoblast lineage cells [50,52,97,98]. Interestingly, tumor cells cultured in vitro with recombinant RANKL increased IL-6 expression in response to RANKL, and similar results were found with a co-culture of mouse primary osteoblasts with breast cancer cells or conditioned media from breast cancer cells [99,100], suggesting that IL-6 and RANKL form a feed-forward loop in bone-DTCs. These studies suggest that breast cancer cells within the bone microenvironment may interact with osteoblast lineage cells to produce cytokines like IL-6 to either promote tumor growth or induce bone resorption. These data are also more consistent with the in vitro 3D study which suggests that IL-6 enhances tumor cell proliferation [96] than the numerous 2D studies that suggest opposing effects on proliferation [87,88,89,90,91,92]. The treatment of mice with anti-IL-6R antibodies resulted in similar cellular growth inhibition in prostate cancer, reduced osteolytic lesions, and a reduction in serum RANKL levels in vivo [7]. Since IL-6 signaling in bone-disseminated tumor cells might be driven by *cis-* or *trans-*IL-6 signaling [27], future studies investigating the efficacy of these neutralizing antibodies on both types of signaling are of interest. Additionally, the shRNA targeting of RANKL in breast and prostate cancer, and shRNA targeting of IL-6 in breast cancer, resulted in smaller osteolytic lesions, reduced bone turnover, and reduced osteoclast numbers in inoculated mice. These data suggest that RANKL secreted by osteoblasts in response to IL-6 from tumor cells contributes to the preservation of RANKL-induced osteoclast activity. Furthermore, tumor cells exposed to osteoblast-derived RANKL increase their IL-6 output [100]. This was corroborated by a separate study which found that RANK (the receptor for RANKL) knockdown in MDA-MB-231 (TNBC) cells reduces osteolytic bone destruction [101]. The IL-6 results were further confirmed in another study by an independent group, where senescent osteoblasts stimulated the production of IL-6, which increased osteoclast number and activity, promoting a metastatic ‘niche’ for breast tumor cells, and bone colonization was reduced following treatment with an IL-6 neutralizing antibody [102]. Taken together, these data indicate a pro-tumorigenic role for IL-6 expressed by bone-DTCs through their interactions with the bone microenvironment. The expression of IL-6 can also be driven by IL27-p28 (IL-30), which has tumor-promoting effects in prostate cancer [103,104] and in breast cancer is enriched for and associated with the TNBC subtype. In breast cancer, the source of IL-30 was stromal leukocytes, and IL-30 stimulated the proliferation of breast cancer cells in a gp130/IL-6R- and STAT1/STAT3-mediated mechanism [105]. These studies were carried out in the context of the primary tumor and not metastatic disease, but these data are consistent with the observed tumor-promoting effects of IL-6 on bone-disseminated tumor cells and it is therefore possible that some of these effects may be mediated through IL-30-driven IL-6 signaling.

*LIF:* The LIF receptor (LIFR) was identified as a breast tumor suppressor by an shRNA screen [74] and shown to function as a breast cancer lung metastasis suppressor by a second laboratory [24]. In SUM159 human breast cancer cells, the knockdown of LIFR dramatically increased the ability of tumor cells to colonize the lungs, while ectopic LIFR expression in 4T1 mouse mammary carcinoma cells significantly reduced the ability of these cells to colonize the lungs [24]. Breast cancer cell lines with low metastatic potential, defined by their lack of colonization of the lung or bone marrow following intravenous inoculation (e.g., MCF7, SUM159, and D2.0R cells), abundantly express LIFR and initiate downstream signals in response to recombinant LIF, but highly metastatic breast cancer cell lines (e.g., MDA-MB-231b, 4T1BM2, and D2A1 cells) do not express a functional LIFR and are unresponsive to recombinant LIF treatment [73], suggesting that the ability of cells to respond to LIF corresponds to their metastatic potential. Interestingly, the restoration of LIFR in highly aggressive MDA-MB-231 cells by treatment with a histone deacetylase inhibitor restores STAT3 signaling downstream of LIF:LIFR, which has been proposed to promote drug resistance by breast cancer cells [106]. 

When evaluating metrics of tumor dormancy, the MCF7 breast cancer cell line has been used by our group and others as a model of tumor dormancy because of its limited growth in the bone microenvironment [71,73,107]. The knockdown of LIFR in MCF7 cells increased invasion, downregulated dormancy genes, and increased osteolytic bone destruction [73]. Furthermore, PTHrP overexpression in MCF7 cells, which effectively enables the cells to exit dormancy in the bone marrow and become aggressively osteolytic [108], also down-regulates LIFR and SOCS3 [73], independent of cAMP signaling [5]. These data are consistent with the role of LIFR as a metastasis suppressor [24]. However, it remains unclear whether LIF in fact drives the metastasis suppressor actions of LIFR, since most data suggest that LIF is tumor-promoting. Several other ligands (OSM and CNTF included) are also able to bind to LIFR, which may mediate the tumor-suppressive actions of LIFR. Recently, the interleukin-like epithelial-mesenchymal transition (EMT) inducer (ILEI) has emerged as a new cytokine that can activate STAT3 and drive both EMT and breast cancer stem cell formation through LIFR [109]. It is important to also note that the acetylation of LIFR on its juxtamembrane domain appears to be responsible for LIF-mediated STAT3 activation and that the phosphorylation of LIFR suppresses LIF signaling [110,111]. Previous work has demonstrated that LIF can induce STAT3 signaling in breast cancer cells [73,106], and STAT3 has been previously identified as a pro-dormancy factor in ER+ breast cancer cells [107] and prevents colonization of the bone by disseminated tumor cells [73]. Therefore, while in vitro treatment with LIF may stimulate tumor cell proliferation, its ability to stimulate STAT3 signaling in the context of the bone microenvironment may still promote tumor dormancy, although the mechanism is unresolved. Further studies will be required to examine the mechanism of action for LIF effects on breast cancer cells in the context of the bone microenvironment. 

The role of LIF signaling in cancer progression appears to be tumor-type dependent, although this does not resolve all of the controversy. In breast cancer, numerous studies point to a tumor-promoting role for LIF. LIF increased the proliferation and colony formation of MCF7 [112] and T47D cells in a dose-dependent manner in vitro, and this effect was reversed when cells were treated with anti-LIF antibodies [113]. LIF also stimulated migration and invasion in ER+ MCF7, T47D, and MDA-MB-231 TNBC cells in vitro using trans-well assays, and the overexpression of LIF in these cell lines increased the number of lung metastases and distant metastases in vivo [114]. However, since these studies made use of the MDA-MB-231 breast cancer cell line, which several groups have shown does not express a functional LIFR [73,106], it is unclear how LIF may stimulate the migration and invasion of these cells in vitro. LIF effects on metastasis were ablated through the shRNA knockdown of LIF in MDA-MB-231 cells [114], but given the absence of a functional LIFR in these cells, the effects observed in vivo would most likely be mediated through paracrine LIF signaling from the tumor cells to the microenvironment. 

In contrast to the pro-tumorigenic effects of LIF identified in MDA-MB-231 TNBC cells, two independent groups have demonstrated that LIF can have a mild inhibitory effect on proliferation in vitro. ER+ MCF7 cells, which do have a functional LIFR [73], displayed significant growth reduction following treatment with exogenous LIF [54,73,115], a decrease in the number of cells in S phase [115], and reduced clonogenic potential [54]. Therefore, LIF treatment on MCF7 cells has been reported to have both positive and negative effects on cellular proliferation in two different clonogenic assays [54,112,113], but the differential effects may have arisen from the types of soft agar used and the chemical make-up of these assays. Other methods have been used to determine the role of LIF on cellular proliferation in vitro, such as XTT assays [73], absolute cell counts [114,115], and flow cytometry [115], but the growth-promoting or inhibitory effects may stem from the limitations of each test. A more standardized approach to in vitro clonogenic assays [116] has been heavily used by a number of groups and may be useful for addressing the discrepancy in previous studies. The differential effects of LIF treatment across several studies also may be due to varying sources and activities of the recombinant cytokine used by each group. Several groups have also reported using a wide range of concentrations, anywhere between 6 and 200 ng/mL, which, when coupled with the different sources of recombinant cytokine, could explain why the results are paradoxical. The mixed outcomes of these studies demonstrate that LIF signaling in breast cancer is controversial and at this time, it is unclear whether this is associated with the hormone receptor status.

*OSM:* Early studies of OSM effects on breast cancer cell proliferation suggested a growth-inhibitory role for this cytokine. Breast cancer cells treated with OSM exhibited reductions in DNA synthesis ((^3^H)thymidine incorporation) in a dose-dependent manner [117], decreased absolute cell counts [115], and a reduction in the number of cells in the S phase [115,118]. In support of these findings, another group has published that OSM inhibits the growth of MCF7 and MDA-MB-231 cells [119], as well as human breast epithelial cells [118,120]. More recently, several studies have focused on OSM’s role in EMT and invasion. Treatment with OSM resulted in the morphological redistribution of β-catenin, enhanced mammosphere formation in T47D and MCF7 cells, suppressed E-cadherin expression, and the increased expression of N-cadherin in MCF7 cells [121], suggesting that OSM promotes EMT; however, these findings have not been tested in vivo. Other studies have pointed to OSM inducing morphological changes necessary to enhance the metastatic characteristics of various breast cancer cells. In the presence of OSM, T47D cells exhibited decreased intercellular contact [122], and increased cellular detachment and invasiveness [123]. In patient data, high OSM expression has also been correlated with decreased patient survival, pointing to its possible role in metastatic disease [124]. In the context of breast cancer bone metastasis, one group has shown that OSM knockdown in 4T1 mouse mammary carcinoma cells reduced spontaneous metastasis to the spine, as assessed by qPCR analysis following orthotopic injections, and less osteolytic bone destruction following intratibial injections [125]. This group also demonstrated that the global knockdown of OSM in Balb/c mice reduced the formation of spontaneous lung metastases [126]. These data suggest that autocrine OSM promotes bone metastasis, and paracrine OSM signaling promotes lung metastasis, but the mechanism by which OSM acts in vivo to stimulate metastasis remains unclear.

While the direct stimulation of OSM on breast cancer cells results in a growth-inhibitory phenotype [115,117,118,119], OSM has also been shown to enhance the invasiveness and metastasis of tumor cells. However, OSM can signal through both LIFR and OSMR to induce downstream signaling [16,29], and the function of OSM:LIFR and OSM:OSMR signaling in breast cancer cells has not been fully explored [122]. Previous studies have shown that STAT3 is abundantly phosphorylated in response to OSM across a number of breast cancer cells lines, with no correlation to their status of LIFR or the metastatic phenotype [73]. Since the status of OSMR on breast cancer cell lines has not been fully elucidated yet, the effects of OSM could be delineated between the expression and functionality of OSMR and LIFR in such cell lines. Interestingly, OSMR expression was associated with shorter recurrence-free survival and overall survival in breast cancer patients [55], suggesting a connection to disease progression in breast cancer. 

*Activation of downstream signaling by the gp130 family:* Upon binding to gp130 and their cytokine-specific receptor on tumor cells, the gp130 ligands are known to activate the JAK/STAT, MAPK/MEK/ERK, and PI3K/AKT signaling pathways [20,21,22,23]. While there has not been a comprehensive comparison of the downstream pathways activated by each ligand in breast cancer, many mechanistic studies have identified STAT3 as the key downstream mediator of IL-6 [127] and OSM [124,128,129] tumor-promoting effects. Signal transduction by IL-6, LIF, and OSM is initiated after dimer formation between the cytokine specific receptors (e.g., IL-6R, LIFR, and OSMR) and gp130, resulting in the phosphorylation of STAT3 by JAK [16,130,131]. Phosphorylated STAT3 undergoes dimerization and translocates into the nucleus, resulting in the transcription of target genes [127]. An extensive body of literature has established a role for STAT3 in the induction of pro-inflammatory cytokines, tumor progression, initiation, metastasis, chemoresistance, and immune evasion [127,132]. STAT3 has also been shown to cross-talk with Ras signaling to further promote the oncogenic transformation of human mammary epithelial cells [133], and recent studies of ovarian cancer have pointed to STAT3’s ability to promote metastasis, chemoresistance, and EMT via MAPK/PI3K/AKT signaling downstream of p53/Ras signaling [134]. 

It is therefore not surprising that the ectopic expression of IL-6 or treatment with recombinant IL-6 in ER+ breast cancer cells significantly increases the expression of EMT-related genes through STAT3, leading to increases in tumor cell proliferation in orthotopic xenograft models [135]. IL-6 has been shown to promote breast cancer metastasis by upregulating C-X-C chemokine receptor type 4 (CXCR4) through c-Jun, STAT3, and nuclear factor kappa B (NF-κB) [136,137], and facilitate angiogenesis through STAT3 by upregulating vascular endothelial growth factor (VEGF), matrix metallopeptidase 9 (MMP9), and basic fibroblast growth factor (bFGF) in the tumor microenvironment [138,139]. OSM induces similar pro-tumorigenic changes in breast cancer cells. Long-term exposure of OSM to human mammary epithelial cells (HMEC) in culture drove EMT changes and resulted in the generation of cancer stem cells (CSC) by inducing STAT3/SMAD3 signaling [140]. This effect was ablated using a TGFβRI inhibitor or the expression of SMAD7 (inhibitor of SMAD3 phosphorylation). This study also noted that several gp130 cytokines (IL-6, LIF, CT-1, and CNTF) were able to significantly increase the CSC population in HMECs, pointing to their potential role as microenvironmental cytokines capable of promoting tumor progression through STAT3. OSM has also been shown to induce IL-6 in a STAT3-dependent manner in ER- breast cancer cells [124]. 

Interestingly, while LIF is known to activate the same downstream signaling pathways as OSM and IL-6, STAT3 appears to play a contradictory role in the context of tumor cell dormancy in the bone. In ER+ MCF7 breast cancer cells, the pharmacological inhibition of MAPK/MEK/ERK or PI3K/AKT signaling had no effect on dormancy markers, while a STAT3 inhibitor reduced pro-dormancy genes [73]. These data were confirmed in vivo, where the knockdown of STAT3 phenocopied the knockdown of LIFR and led to tumor cell exit from dormancy. STAT3 is up-regulated in dormant tumor cells and was one of only six genes that was highly expressed in ER+ breast cancer cell lines with higher dormancy scores [107]. The downstream mechanism for these effects remains unclear, but is intriguing as it has been observed across multiple independent dormancy studies. These data suggest that the inhibition of STAT3 in the primary site is critical to reducing tumor cell growth, but in distant metastatic sites, such as the bone marrow, STAT3 inactivation could lead to the awakening of dormant tumor cells. In contrast to this, the small molecule inhibitor EC359, which has been shown to directly interact with LIFR to block its interactions with LIF, OSM, CNTF, and CT-1, reduced LIFR-mediated activation of multiple gene targets, STAT3 activity, and downstream target genes, and suppressed TNBC xenograft and PDX tumor growth in vivo [141]. While ER+ breast cancer cell lines were used in the initial screens for LIF and LIFR in these studies, functional studies were all carried out in TNBC cell lines, so it is unclear whether EC359 would have had similar effects on ER+ tumor progression in vivo. 

While the predominant signaling pathway activated by the gp130 cytokines is the JAK-STAT signaling axis, OSM has been shown to suppress ER protein and mRNA expression in ER+ breast cancer cells through the MAPK-ERK pathway [55]. The MAPK inhibitor U0126 blocked morphological changes in ER+ breast cancer cell lines, confirming MAPK as a downstream mediator of the pro-migratory phenotype induced by OSM. In combination with the STAT3 studies above, this points to OSM activating multiple signaling pathways to promote breast cancer progression. While OSM, LIF, and IL-6 can induce STAT3, AKT, and ERK signaling in breast cancer cells, in the absence of studies using combinations of STAT3, AKT, and ERK inhibitors, it is difficult to determine which is the dominant downstream mediator. Current literature suggests that LIF and IL-6 elicit functions primarily through STAT3 activation, while OSM may act through both STAT3 and ERK. 

## 5. IL-6 Cytokines and Cancer Stem Cells 

Recently, several groups have proposed that tumor cells that reside in a dormant state do so through the adoption of a cancer stem cell (CSC) phenotype [57,142]. It has been proposed that CSCs are a subset of cancer cells that undergo self-renewal [143], are responsible for tumor initiation, progression, and metastases [143,144,145,146], and persist long-term [142], but this has not been well-studied with regards to bone metastasis. CSC populations have also been associated with a poor prognosis and increased resistance to chemo/radio-therapies [147,148,149]. Certain cytokines are already known to stimulate the expression of CSC features, including transforming growth factor-beta (TGF-β) by activating Wingless (Wnt) signaling in breast cancer cells [150]. The self-renewal properties of CSCs can also be regulated by a network of regulatory and signaling pathways, such as Notch [151], Hedgehog [152,153], TGF-β [154], estrogen/progesterone receptor (ER/PR) [155], epidermal growth factor/receptor (EGF/EGFR) [156], and LIF [157]. The overexpression of several of these signaling pathways is known to increase the stem cell and CSC pool [152,158]. Of these ligands, LIF has known functions as a pro-stemness factor by maintaining pluripotency in mouse embryonic stem cells [159,160]. For the culture of mouse embryonic stem cells (mESCs), LIF and other gp130 cytokines, such as OSM, CNTF, and CT-1, can enable self-renewal and promote stem-ness by activating pluripotency-associated genes through STAT3 [161,162,163,164,165,166,167]. In pancreatic cancer, a blockade of LIF:LIFR:STAT3 signaling resulted in a decrease in the expression of CSC-associated markers (CD133, CD24, and CD44), a reduction in tumor initiation and formation, and an overall less aggressive phenotype [168]. In addition, long-term stimulation by OSM on transformed-human mammary epithelial cells (HMECs) resulted in an increase in CD44^High^/CD24^Low^ expression and upregulation of CSC/EMT-associated genes, and promoted stem cell plasticity both in vitro and in vivo [140,169], suggesting that long-term OSM exposure may promote a CSC phenotype.

It has been previously established that LIFR expression on breast cancer cells promotes tumor dormancy in the bone marrow and that several of the gp130 cytokines (LIF, OSM, and CNTF) are able to signal through LIFR. Because LIF, OSM, and CNTF are present in the bone marrow, bone-DTCs may receive these signals in the endosteal niche to remain in a dormant state; however, given that these cytokines can also promote stemness, it will be important to determine whether LIF, OSM, and CNTF induce dormancy by promoting a CSC phenotype in which the cells are more quiescent, but have the potential for self-renewal. 

## 6. Clinical Implications

The clinical importance of LIF, OSM, and CNTF in breast cancer has yet to be established. While studies have confirmed that the loss of LIFR:STAT3 signaling is associated with reduced breast cancer patient survival [24,73], to date, there are no clinical studies that have rigorously tested these cytokines and their receptors in the oncology setting. In contrast, given the pro-tumorigenic role for IL-6 in vivo and decades of work establishing its ability to promote tumor cell survival, invasion, and metastasis, there has been great interest in the therapeutic targeting of IL-6/JAK/STAT signaling [127]. lL-6 levels are elevated in patients across multiple tumor types, including breast cancer [170,171,172,173,174,175,176,177,178,179,180], and mechanistically, IL-6 has been shown to promote breast cancer stem cell renewal through Notch3 signaling [138] and distant metastasis of breast cancer cells to the lung through STAT3 signaling [181] and to the bone through RANKL signaling [100]. Therefore, there are multiple antibodies that have been developed to inhibit IL-6 signaling, including siltuximab (targets IL-6) and tocilizumab (targets the IL-6R). Siltuximab is currently FDA-approved for the treatment of multicentric Castleman disease [182] and is being used in multiple clinical trials for the treatment of hematologic cancers and solid tumors [127]. Tociluzimab is currently FDA-approved for the treatment of rheumatoid arthritis and systemic juvenile idiopathic arthritis, and is in clinical trials for the treatment of metastatic HER+ breast cancer (NCT03135171). Therefore, there are significant clinical efforts to therapeutically target IL-6 to prevent the progression and spread of solid and hematologic cancers. 

Interestingly, IL-6 is now being targeted to mitigate the effects of some of the most recent, novel therapies to be adopted in the clinical oncology setting. The most prevalent adverse event following chimeric antigen receptor (CAR) T cell therapy, which uses manipulated T cells to target tumor antigens, is the rapid onset of immune activation known as the ‘cytokine storm’ or cytokine release syndrome (CRS) [183,184,185]. In CRS, activated T cells produce cytokines, including monocyte chemoattractant protein-1 (MCP-1), IL-8, IL-10, IL-2, interferon gamma (IFNγ), IL-6, and IL-6R [186,187,188,189,190], which recruit more immune cells and lead to a highly inflammatory state. One study demonstrated that sIL-6R and sgp130 serum levels are strongly associated with the development of severe CRS in patients, but the risk of developing CRS was significantly reduced when patients were started on tocilizumab [191]. Tocilizumab was recently approved by the FDA to treat CRS and has become the standard of care following CAR T-cell infusion [192,193,194]. It is worth noting that CRS is not exclusive to CAR T therapy and can also be triggered following treatment with immune checkpoint inhibitors [195]. 

Given that CRS can range from mild to life-threatening, there have been substantial efforts to predict those patients that may develop CRS following treatment with CAR T or immune checkpoint therapies through the use of scalable multiplex assays that focus on the Fc gamma receptors (FcγRs) [196], since therapeutic IgG monoclonal antibodies elicit downstream responses through interactions with FcγRs [197]. These multiplex cytokine assays have been used in vitro to identify Fc gamma receptor polymorphisms that predict IFNγ release following treatment with Campath-1H human IgG1, which can induce CRS. The genotyping of whole blood from 11–12 healthy human donors identified the FcγRIIIa-V158F polymorphism, which predicted the magnitude of IFNγ release following treatment with a Campath-1H homolog [198]. In a separate study analyzing 271 whole blood samples from healthy human donors, patients homozygous for the *FCGR2A-131H* and *FCGR3A-158V* alleles demonstrated elevated IFNγ production in response to Campath [199]. The utility of using these polymorphisms to predict the response rate to drugs has been demonstrated with the FCGR2B polymorphism, which corresponded to a reduced response rate to rituximab, which targets CD20 in B cell malignancies [200]. Importantly, multiplex assays that target the FcγR locus are scalable for use in clinical trials [196], and may be instrumental in predicting and ultimately mitigating CRS in cancer patients receiving immune targeted therapies. Other efforts to mitigate CRS and potential immune therapy-related toxicities include an engineering approach to reduce glycosylation and increase sialylation of the Fc domain of the antibodies [201], including trastuzumab, which targets HER2 and is FDA-approved for the treatment of HER2+ breast cancer [202,203].

## 7. Conclusions

In summary, the gp130 cytokine family regulates a wide range of processes that affect bone remodeling, cancer pathogenesis, and metastasis through paracrine and autocrine mechanisms. Despite the body of literature defining the numerous roles of the gp130 cytokine family members, there are still mechanisms of action that remain unknown, particularly with regards to the contradictory effects of OSM and LIF on breast cancer cells. These cytokines can complex with LIFR on breast cancer cells, which promotes dormancy in bone-disseminated tumor cells, and is therefore highly relevant to the pathogenesis of bone-disseminated breast cancer cells. The cytokines capable of binding to LIFR and the other cytokine-specific receptors in the gp130 family are produced by bone-resident osteoblast and osteoclast-lineage cells and are therefore likely to reach bone-disseminated tumor cells and alter their behavior and signaling (Figure 2). Likewise, since bone marrow-resident cells express the gp130 subunit and many of the cytokine-specific receptors (e.g., LIFR and OSMR), the tumor cell production of these cytokines may also remodel the bone marrow microenvironment to make it more permissive for tumor colonization or dormancy. The role of these cytokines in breast cancer bone colonization and bone metastasis in particular remains unclear, but a better understanding of the similarities and differences in the signaling pathways and behavior of breast cancer cells in response to each of these cytokines would benefit the bone metastasis field. Increased knowledge of the molecular mechanisms of this cytokine family may lead to the development of new therapeutic targets to prevent tumor dissemination to and progression in the bone. 

## Figures and Tables

**Figure 1 cancers-12-00326-f001:**
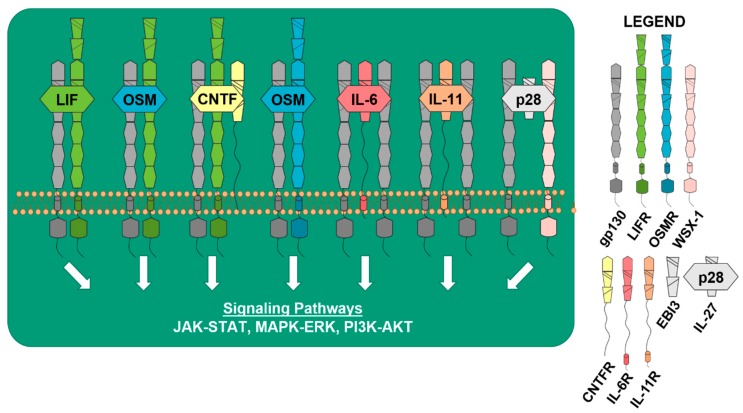
gp130 cytokines and receptors activate downstream signaling pathways. Receptors: dark gray = glycoprotein130 (gp130) co-receptor, green = leukemia inhibitory factor (LIF) receptor (LIFR), blue = oncostatin M (OSM) receptor (OSMR), light pink = WSX-1 (interleukin 27 receptor subunit alpha), yellow = ciliary neurotrophic factor (CNTF) receptor (CNTFR), dark pink = interleukin-6 (IL-6) receptor (IL-6R), orange = interleukin-11 (IL-11) receptor (IL-11R), light gray = Epstein-Barr virus induced 3 (EBI3), and EBI3+IL-27p28 (IL-30) = interleukin-27 (IL-27). LIF, OSM, CNTF, IL-6, Il-11, and IL-27 bind to their cytokine-specific receptors to activate major downstream signaling pathways: the Janus-activated kinase (JAK)–signal transducer and activator of transcription (STAT) pathway, the Ras-Raf mitogen-activated protein kinase (MAPK and MEK/ERK) signaling cascade, and the phosphatidylinositol 3-kinase-dependent (PI3K/AKT) pathway.

**Figure 2 cancers-12-00326-f002:**
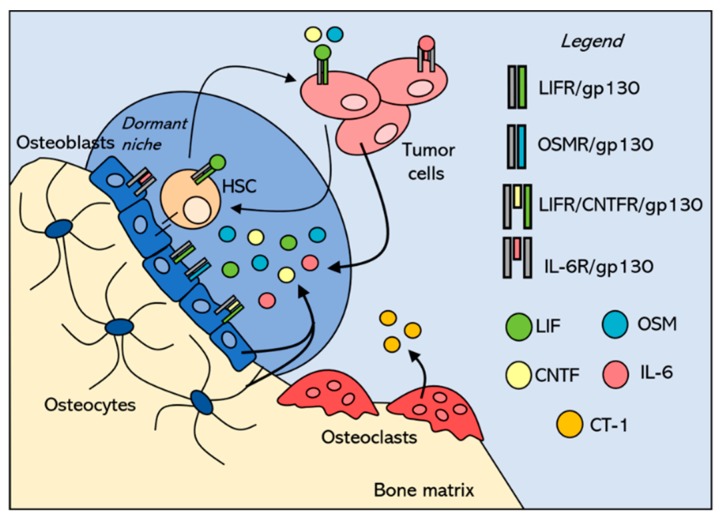
Bone-disseminated tumor cells compete with hematopoietic stem cells (HSCs) in the endosteal niche, where they encounter pro-dormancy cytokines in the microenvironment. Tumor cells that disseminate into the bone marrow are proposed to compete with HSCs for the endosteal niche, which maintains dormancy through cell–cell interactions and secreted factors, including the gp130 cytokines. These cytokines normally send pro-dormancy signals to the HSCs to maintain their quiescence, and when tumor cells compete for this niche, are likely to encounter the same cytokine milieu. Both HSCs and breast cancer cells express LIFR, although LIFR is markedly down-regulated in more aggressive breast cancer cells This suggests that both HSCs and breast cancer cells are capable of responding to LIF, OSM, and CNTF secreted within the bone marrow microenvironment. The sources of these cytokines in the pro-dormancy niche are bone-lining osteoblasts and osteocytes embedded within the bone matrix. Osteoclasts do not express most of the gp130 cytokines, but do express CT-1, which can also bind to LIFR. It is unclear how this might contribute to the pro-dormancy niche along the quiescent osteoblast-lined surface. LIF = leukemia inhibitory factor, OSM = oncostatin M, CNTF = ciliary neurotrophic factor, CT-1 = cardiotrophin-1, IL-6 = interleukin-6, gp130 = glycoprotein130 co-receptor, LIFR = leukemia inhibitory factor (LIF) receptor, OSMR = oncostatin M (OSM) receptor, CNTFR = ciliary neurotrophic factor (CNTF) receptor, and IL-6R = interleukin-6 (IL-6) receptor.

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
