# Peer review of "GP130 Cytokines in Breast Cancer and Bone"

_cancers, 2020, doi:10.3390/cancers12020326_

Round 1
Reviewer 1 Report
Very interesting review. Well written and documented. I would suggest more insights on the downstream receptor cell signaling. The authors could speculate further on the similarities and differences between gp130-cytokines, particularly OSM and LIF in breast cancer metastasis.
Author Response
Very interesting review. Well written and documented. I would suggest more insights on the downstream receptor cell signaling. The authors could speculate further on the similarities and differences between gp130-cytokines, particularly OSM and LIF in breast cancer metastasis.
Response: We have now added a section discussing the downstream receptor signaling and the similarities and differences between the gp130 cytokines in section 4, pages 9-10, lines 418-475.
Reviewer 2 Report
Omokehinde and Rachelle Johnson provide a comprehensive analysis of the role of gp130 cytokines in bone physiology and tumor-bone interaction, with essential implications for breast cancer invasion and metastasis. The emphasis of the review is mainly on breast cancer, but the authors often include information on gp130 cytokines in other cancer cell types.
Overall, the subject is of interest for cancer research, the review is well written and the literature overview well documented. For these reasons, the paper can be published in this current form
Author Response
Omokehinde and Rachelle Johnson provide a comprehensive analysis of the role of gp130 cytokines in bone physiology and tumor-bone interaction, with essential implications for breast cancer invasion and metastasis. The emphasis of the review is mainly on breast cancer, but the authors often included information on gp130 cytokines in other cancer cell types.
Overall, the subject is of interest for cancer research, the review is well written and the literature overview is well documented. For these reasons, the paper can be published in this current form.
Response: Thank you. We appreciate these comments.
Reviewer 3 Report
The authors have reviewed the literature on IL-6/gp130 co-receptor and related family cytokines/their receptor axes for understanding the tumor biology of breast cancer cells and their skeletal remodeling and cancer progression both within and outside the bone microenvironment. The signal transduction of interleukin-6 (IL-6) cytokine family, its use of the glycoprotein 130 (gp130) co-receptor, includes other cytokines such as interleukin-11 (IL-11), leukemia inhibitory factor (LIF), oncostatin M 17 (OSM), ciliary neurotrophic factor (CNTF), and cardiotrophin-1 (CT-1).
These cytokine/ their receptor axes are quite complex, and all necessary pros-cons of their biological implications need to be addressed in the article. I have following major and minor comments that need to be addressed with a supportive assessment with recent literature .
1, IL-6/its receptor and go130 co-receptor axis is a powerful axis and has several tumor biological implications after initiating signal transduction. It also looks on many type 1 and type 2 cytokine pathways. This information is missing in the article, which should be adequately discussed.
IL-6 is also critically involved in cytokine storm that is known for inducing severe adverse effects in vital and non-vital organs in pre-clinical and clinical studies not only in breast cancer subjects but also in other types of cancer subjects. The authors should look into a variety of toxicity studies induced by these cytokines, the evolvement of different therapies to control of IL-6 related toxicities and summarize with supportive recent literature. How these cytokines are secreted either autocrine or paracrine or both in a real-time manner in experimental models to understand severity of disease pathogenesis in vivo in cancer patients? The authors should look into the observations made in many recent studies that have employed detecting and measuring many cytokines using multiplex detection technologies and their impact on disease progression in animal studies and their correlation in investigational human studies. Similarly, they could have provided more information briefly to address the signal transduction related to these cytokines and other critical issues on other type of solid tumors that may be either hormone dependent or independent. Importantly, the authors should provide information on recent advances made by several investigators to control the adverse effects of this important class of cytokines either by biological agents, cellular or gene therapy based products and their effectiveness in controlling disease progression and metastasis. These issues should be elaborated in a separate paragraph. The authors should search the recent literature and update the article as only 11% of the cited references are after 2015.
Author Response
The authors have reviewed the literature on IL-6/gp130 co-receptor and related family cytokines/their receptor axes for understanding the tumor biology of breast cancer cells and their skeletal remodeling and cancer progression both within and outside the bone microenvironment. The signal transduction of interleukin-6 (IL-6) cytokine family, its use of the glycoprotein 130 (gp130) co-receptor, includes other cytokines such as interleukin-11 (IL-11), leukemia inhibitory factor (LIF), oncostatin M 17 (OSM), ciliary neurotrophic factor (CNTF), and cardiotrophin-1 (CT-1).
These cytokine/ their receptor axes are quite complex, and all necessary pros-cons of their biological implications need to be addressed in the article. I have following major and minor comments that need to be addressed with a supportive assessment with recent literature .
IL-6/its receptor and go130 co-receptor axis is a powerful axis and has several tumor biological implications after initiating signal transduction. It also looks on many type 1 and type 2 cytokine pathways. This information is missing in the article, which should be adequately discussed.
Response: We now discuss IL-6 in more depth in a new section (section 6, “Clinical implications”) on page 11, lines 506-523.
IL-6 is also critically involved in cytokine storm that is known for inducing severe adverse effects in vital and non-vital organs in pre-clinical and clinical studies not only in breast cancer subjects but also in other types of cancer subjects. The authors should look into a variety of toxicity studies induced by these cytokines, the evolvement of different therapies to control of IL-6 related toxicities and summarize with supportive recent literature. How these cytokines are secreted either autocrine or paracrine or both in a real-time manner in experimental models to understand severity of disease pathogenesis in vivo in cancer patients? The authors should look into the observations made in many recent studies that have employed detecting and measuring many cytokines using multiplex detection technologies and their impact on disease progression in animal studies and their correlation in investigational human studies.
Response: We have now added a discussion of the role of IL-6 in the adverse events induced by novel tumor-targeted therapies in section 6, pages 11-12, lines 524-536.
Similarly, they could have provided more information briefly to address the signal transduction related to these cytokines and other critical issues on other type of solid tumors that may be either hormone dependent or independent.
Response: We have now added a section discussing the downstream signal transduction in section 4, pages 9-10, lines 418-475 and how this relates to hormone receptor status. We have also added additional text throughout that clarifies the hormone receptor status of the breast cancer cell lines discussed and how this aligns with the effects of each ligand.
Importantly, the authors should provide information on recent advances made by several investigators to control the adverse effects of this important class of cytokines either by biological agents, cellular or gene therapy based products and their effectiveness in controlling disease progression and metastasis. These issues should be elaborated in a separate paragraph.
Response: We now address the control of adverse effects induced by IL-6 cytokines in the clinical oncology setting in section 6, page 12, lines 537-555.
The authors should search the recent literature and update the article as only 11% of the cited references are after 2015.
Response: We appreciate the comment and agree that it is important to highlight the latest literature in reviews. Much of the work that determined the role for the gp130 cytokines in bone occurred prior to 2015, and many of the studies that examined the effects of the gp130 cytokines on breast cancer cells were published several decades ago. The focus of this review was, in part, to lay out the conflicting data for these cytokines that have been published over the course of several decades and examine the similarities, differences, and inconsistencies. Thus, the nature of this review pulls in quite a few historic references. We have endeavored to add additional recent publications, particularly with the addition of the Clinical implications section (pages 11-12, lines 506-555). In this revision we have added 71 new references, and of these, 31 were published between 2015-2020.
Reviewer 4 Report
In this paper, Omokehinde and Johnson, focused on GP130 cytokines in breast cancer and bone. They described effects of p130 in bone remodeling and breast cancer. This review is rather comprehensive and well written. Nevertheless, some points have to be described.
Major comment:
The effects of cytokine and gp130 are not associated with molecular portrait of breast cancer but we well-know that they are association between hormonal status and expression. Moreover, SERM modulated gp130 and cytokine expression. So, the authors have to report, at least, the correlation between ER, TBNC or HER2 and GP130 and cytokine. Furthermore, when they cited cell lines it will be better to say whether they are ER, TBNC or HER2. The terms less aggressiveness or low metastatic, I think, are not sufficient to characterize these cell lines. MCF-7 were isolated from pleural effusion so they are metastatic and the patient was died from her cancer so are they less aggressive? Moreover, Ras status is different among these cell lines and we know that ras and STAT3 signaling are linked (Liang et al, Oncogenesis, 2019). So the authors have to nuanced the choice of the cell lines and for each example they have to be more “precise” and describe which cell characteristics are probably responsible to cell response (ER status, p53 status, ras mutation, HER2, TN status…)
Minor comments
Line 50. They authors described effect of Il6st KO in mice in term of lethality. But I think that they have to add for instance shin et al work on osteoblast (Shin et al, Endocrinology. 2004 Mar;145(3):1376-85) and/or Kawasaki K et al (Endocrinology. 1997 Nov;138(11):4959-65).
Line 69. Figure 1, IL27 and IL11 are also described to acts throught GP130 recruitment so the authors have to complete the cartoon. Moreover, IL6 expression is driven by IL27p28 and support TBNC growth (Airoldi et al, Cancer Res. 2016).
Line 104 please give the detail of HCS abbreviation here and not line 106.
Author Response
In this paper, Omokehinde and Johnson, focused on GP130 cytokines in breast cancer and bone. They described effects of p130 in bone remodeling and breast cancer. This review is rather comprehensive and well written. Nevertheless, some points have to be described.
Major comment:
The effects of cytokine and gp130 are not associated with molecular portrait of breast cancer but we well-know that they are association between hormonal status and expression. Moreover, SERM modulated gp130 and cytokine expression. So, the authors have to report, at least, the correlation between ER, TBNC or HER2 and GP130 and cytokine. Furthermore, when they cited cell lines it will be better to say whether they are ER, TBNC or HER2. The terms less aggressiveness or low metastatic, I think, are not sufficient to characterize these cell lines. MCF-7 were isolated from pleural effusion so they are metastatic and the patient was died from her cancer so are they less aggressive? Moreover, Ras status is different among these cell lines and we know that ras and STAT3 signaling are linked (Liang et al, Oncogenesis, 2019). So the authors have to nuanced the choice of the cell lines and for each example they have to be more “precise” and describe which cell characteristics are probably responsible to cell response (ER status, p53 status, ras mutation, HER2, TN status…)
Response: We have now added discussion of hormone receptor status and the known interactions of estrogen receptor with the gp130 cytokines and receptors, as well as the effects of SERMs on the gp130 cytokine family in section 4, pages 5-6, lines 211-251. We have also provided more detail regarding hormone receptor status for each cell line discussed throughout the manuscript and also discuss the downstream signal transduction for the cytokines in section 4, pages 9-10, lines 418-475. We have also clarified what is meant by metastatic potential on page 8, lines 326-327, which we define as the ability of the cell line to colonize the lung or bone following intravenous inoculation.
Minor comments:
Line 50. They authors described effect of Il6st KO in mice in term of lethality. But I think that they have to add for instance shin et al work on osteoblast (Shin et al, Endocrinology. 2004 Mar;145(3):1376-85) and/or Kawasaki K et al (Endocrinology. 1997 Nov;138(11):4959-65).
Response: The work described by Shin et al. and Kawasaki et al. has now been added to section 1, page 2, lines 51-55.
Line 69. Figure 1, IL27 and IL11 are also described to acts throught GP130 recruitment so the authors have to complete the cartoon. Moreover, IL6 expression is driven by IL27p28 and support TBNC growth (Airoldi et al, Cancer Res. 2016).
Response: We have now added these cytokines to Figure 1 (page 2) and now discuss IL-27p28 in TNBC in section 4, page 7, lines 313-320.
Line 104 please give the detail of HCS abbreviation here and not line 106.
Response: This has now been corrected.
Round 2
Reviewer 3 Report
I have no additional comments. The revised version needs to be spell checked.